# Evaluating the Presence of Lycopene-Enriched Extracts from Tomato on Topical Emulsions: Physico-Chemical Characterization and Sensory Analysis

Ana Costa [1,*,†], Marta Marques [2,†], Franca Congiu [1], Alexandre Paiva [2], Pedro Simões [2], António Ferreira [3], Maria Rosário Bronze [1,3], Joana Marto [1], Helena Margarida Ribeiro [1,*] and Sandra Simões [1]

1   iMed.ULisboa, Faculty of Pharmacy, Universidade de Lisboa, Av. Prof. Gama Pinto, 1649-003 Lisboa, Portugal; franca.c4789@gmail.com (F.C.); mrbronze@ff.ulisboa.pt (M.R.B.); jmmarto@ff.ulisboa.pt (J.M.); ssimoes@ff.ulisboa.pt (S.S.)
2   LAQV-REQUIMTE, Departamento de Química, Faculdade de Ciências e Tecnologia, Universidade NOVA de Lisboa, Quinta da Torre, 2829-516 Caparica, Portugal; martadiasmarques@gmail.com (M.M.); abp08838@fct.unl.pt (A.P.); pcs@fct.unl.pt (P.S.)
3   IBET, Instituto de Biologia Experimental e Tecnológica, Av. da República, Estação Agronómica, Apartado 12, 2780-901 Oeiras, Portugal; antoniof@ibet.pt
*   Correspondence: acosta@farm-id.pt (A.C.); hribeiro@campus.ul.pt (H.M.R.)
†   The two authors contributed equally to this work.

**Abstract:** One of the new trends of personal care industry is the use of organic ingredients derived from nature, in particular, from food-processing residues with proven efficacy. Lycopene is a carotenoid responsible for the red color of several fruits, namely tomato, whose antioxidant and photoprotective effects have been studied. Methods: Lycopene-enriched extracts (LEE) were obtained from tomato waste using supercritical $CO_2$ extraction, incorporated in microemulsions and macroemulsions for topical use, and characterized through GC-MS for the identification of volatile compounds. The aim of this work was to evaluate the impact of the presence of lycopene-enriched extracts developed emulsions through the identification of volatile compounds and by a sensory analysis to assess the odor and color perception and the acceptability of such semi-solid systems as cosmetic products. Results: Volatile compounds were identified in the extract and in the formulations containing the extract. Preliminary data show that the odor of both LEE-loaded microemulsions and LEE-loaded macroemulsions was classified as undefined. The information about the composition did not modify the odor perception but increased the acceptability of some cosmetic products. LEE conferred a yellowish color to formulations, and the information about the formulation composition increased the likelihood of different cosmetic products with this color being bought. Conclusions: The commercialization of personal care products does not only dependent on the associated organoleptic properties, but is influenced by the information about the composition, namely by the presence of an antioxidant compound.

**Keywords:** antioxidant skin supplementation; lycopene extracts; supercritical fluid extraction methods; personal care; sensory analyses

## 1. Introduction

There is an increasing interest and a consuming market growing on natural products named as "green" [1,2]. The search for personal care products formulated with natural ingredients has increased and, currently, there is a consumer preference on products scented with naturally sourced ingredients [3]. In parallel, convergence of food and cosmetics is a major trend in the recent years [4]. Sustainable extraction of natural products from food using innovative technologies constitutes an answer to develop natural fragrant ingredients [5] but also to create products enriched in food-derived bioactives [1]. In fact,

recovering of biological active components from food by-products have gained increasing interest for nutraceutical and cosmetic application since such products represent an important source of antioxidant components [6]. An important example of application is lycopene, a carotenoid that can be extracted from tomato that have gained major significance over the past years in human nutrition [7]. Processed tomato industry generates a large amount of waste (including peel and seeds) possible to be used to generate added-value compounds. Figure 1 illustrates the sustainable valorization of tomato processing industry by-products and the incorporation of waste-derived extracts on both nutritional supplements and cosmetic formulations.

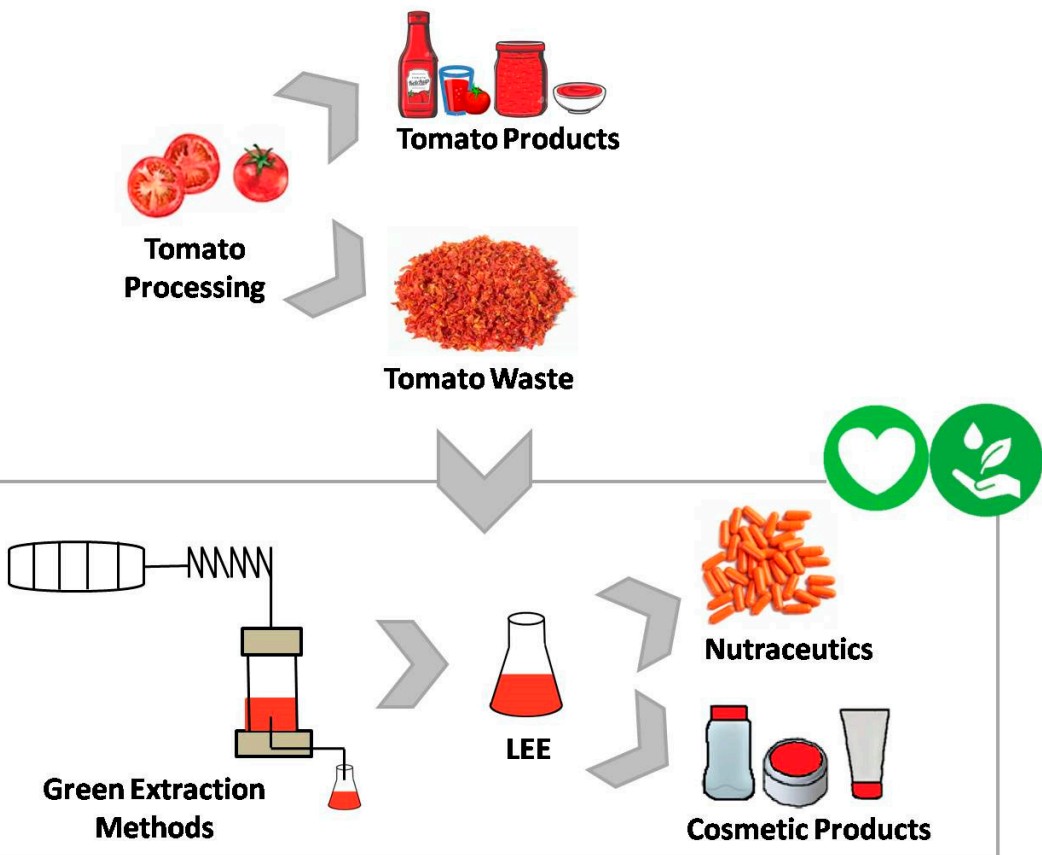

**Figure 1.** Schematic representation of lycopene extraction from tomato processing waste and its application on new products. LEE: Lycopene-enriched extracts.

In the case of nutraceutical products, the presence of an odor associated with a food product does not raise the same acceptability problems as a cosmetic product to be used on the face or body. In fact, the presence of a tomato odor in a cosmetic product can lead to acceptability problems and the good acceptance of a product's odor can be subjectively altered if the consumer is aware of its beneficial properties once applied to the skin.

There are several marketed products claiming a tomato based-origin and the tomato's beneficial properties [8–13] and a tomato scent does not appear as being an acceptability burden for these products. In fact, to the best of our knowledge a tomato scent acceptability study in cosmetics has never been reported. Several studies have based their aim on the efficacy of lycopene, mostly combined with other bioactives, on skin ageing or skin photoprotection. The oral consumption of a mixture of soy isoflavones, lycopene, vitamin C, vitamin E and fish oil, clinically demonstrated to improve the depth of facial wrinkles and to increase the deposition of new collagen fibers in the dermis, following long-term use [14]. The possibility of using plant derived bioactives in sunscreens to protect the skin against sun light exposure damage has been hypothesized [15]; however, the number of studies

including molecules like lycopene in topical forms is very scarce [1]. Apart from lycopene extracted from fruits or food wastes, presenting antioxidant activity, tomato cultured stem cells were developed as a new cosmetic active ingredient and tested for the protection of skin cells towards heavy metal toxicity [16]. Cosmetic ingredients are not odor-free and the cosmetic composition can modify the sensory attributes and influence the consumer acceptance [17]. This means that the final odor of formulations is affected by the qualitative and quantitative composition of each product. It is also known that consumer's response to a cosmetic product is not only based on its efficacy but also on how its attributes are perceived, namely the odor and the color [17].

In this work, a sensorial analysis (odor and the color) of microemulsions and macroemulsions, unloaded and loaded with lycopene-enriched extract (LEE) collected from tomato waste (seeds and skin residues) through supercritical fluid extraction (SFE) was performed. Since, LEE contains several volatile compounds that confer a particular odor to each formulation, this work also aimed to identify the compounds that are responsible for the odor of LEE-loaded formulations.

Sensory analysis in personal care products is an integrated multidimensional measure being able to evaluate how many consumers like or dislike a product, and to detect features that could not be detected by analytical procedures [18]. In this study the sensory analysis was based on two main parameters directly linked to the incorporation of LEE in the formulations: the odor and the color.

Moreover, the characteristic odor of tomato is given by a combination of at least 16 aroma compounds, being the aromatic compounds, aldehydes, and nonpolar alcohols, those with odor potency responsible for tomato flavor [19,20]. In this work gas chromatographic retention indices were determined to identify the volatile compounds of LEE that are responsible for the odor of LEE-loaded formulations. Formulations without LEE were used as a control.

## 2. Materials and Methods

### 2.1. Solvents and Chemicals

Tomato waste collected from *Solanum lycopersicum* crops was provided by Italagro S.A (Castanheira do Ribatejo, Portugal). Triglycerides of medium-chain and glyceryl caprylate/caprate were kindly provided by IOI Oleochemical (Hamburg, Germany). Glycol was bought from PanReac AppliChem (Barcelona, Spain). Polyoxylglycerides were kindly provided by Gattefossé (Saint-Priest, France). Phenoxyethyl caprylate and O/W sugar-based emulsifier were offered by Evonik (Essen, Germany), while the thickening–stabilizing–texturizing were kindly supplied by Seppic (Courbevoie, France). Sorbitan esters were bought from Croda (Barcelona, Spain). Polysorbate, sodium hydroxide, alkane standard solution $C_8$-$C_{20}$ and the FAME C16–C18 standard were purchased from Sigma-Aldrich (St. Louis, MO, USA). The preservative was purchased by Lonza (Basel, Switzerland). All other reagents and solvents used in the present study were of analytical grade and purchased from available suppliers.

### 2.2. Chemical Analysis of Tomato Waste

Tomato waste was lyophilized, milled and stored at $-20$ °C protected from light. The water content of tomato waste was determined by a thermogravimetric balance (Kern DAB 100-3) at 105 °C. Total lipid content was measured by performing a Soxhlet extraction with n-hexane. Ca. 2 g of tomato waste was extracted with 70 mL of n-hexane for 3 h. At the end of the extraction, the solvent was evaporated. The residue was dried overnight at 40 °C, to remove traces of solvent, and weighed.

### 2.3. Supercritical $CO_2$ Extraction of Tomato Waste

SFE of tomato waste was carried out in a high-pressure apparatus. The respective schematic diagram is shown in Figure 2. Gaseous $CO_2$ was taken from a cylinder, liquefied in a cooling bath, compressed to the desired extraction pressure by means of a pneumatic

pump (Williams P250V300), and then heated to the desired temperature, by passing through a high-pressure tube heated by a heating tape. Supercritical carbon dioxide flowed upwards through a packed bed of tomato waste contained in an extraction vessel of ca. 250 cm³ of capacity (570 mm length, 24 mm i.d.; from HiP), at the desired flow rate, measured by a mass flow meter (Rheonik RHM 007). Around 40 g of tomato waste were used per run, dispersed in a bed of glass spheres, and placed in the extraction vessel between two porous metallic plates. Temperature in the extractor was maintained by means of an electrical resistance and controlled by a series of thermocouples connected to a digital controller. The extraction pressure was controlled by means of a back-pressure regulator valve, BPR, (Tescom Europe, model 26-1700, Selmsdorf, Germany) where depressurization of the $CO_2$ flow stream exiting the extraction vessel took place. The extracted substances were precipitated and collected in an electrically heated high-pressure separator (Swagelok 316L-HDF4-500). $CO_2$ was then re-circulated back to the extractor.

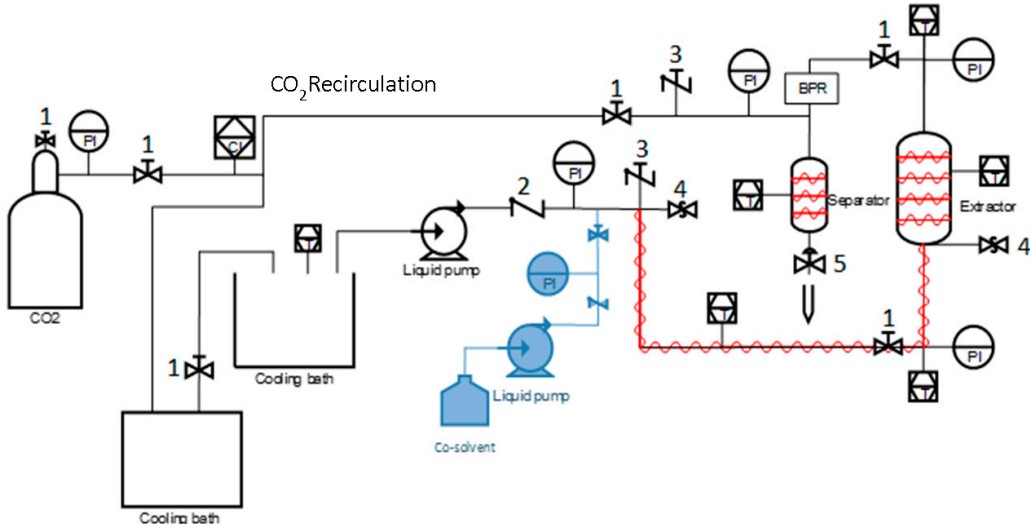

**Figure 2.** Schematic diagram of the SFE apparatus. 1—Needle valve; 2—Check valve; 3—Safety rupture disc; 4—Relief valve; 5—Sample collection valve. BPR—Back pressure regulator.

Based on preliminary experiments not reported here, SFE conditions were set at 500 bar and 60 °C, whereas separation of extract from $CO_2$ was achieved at 50 bar and 50 °C. The $CO_2$ mass flow was set at 8 g/min. Extracted oil was periodically removed from the separator along the run and weighted in order to follow the extraction kinetics. The experiment was ended when a constant weight of extract was achieved

### 2.4. Fatty Acid Analysis of Extracted Oil

The tomato waste oil obtained by Soxhlet extraction and the SFE extracts were analyzed by gas chromatography to determine their fatty acid profile following AOAC method [21]. Fatty acid methyl esters (FAMEs) were prepared and then analyzed in a Thermo Scientific gas chromatograph (Trace ULTRA Series) with an autosampler and flame ionization detector (GC-FID). A Phenomenex ZB 5HT Inferno (30 m × 0.32 mm ID) coated with a 0.10 μm thickness film of 5% dimethyl−95% dimethylpolysiloxane was used for separation. Injections were made in split–splitless mode (SSL), using hydrogen as the carrier gas (1 mL/min). All the data was processed with software Chrom-Card. The FAME were identified by comparison of their retention times with those of chromatographic standards (Sigma Aldrich, St. Louis, MO, USA).

### 2.5. Preparation and Characterization of Microemulsions

For the preparation of ME the aqueous phase was prepared by forming a blend of surfactant (polyoxylglycerides, 25%), co-surfactant (glycol, 25%) and water (40%). The aqueous phase was posteriorly added to the oily phase (glyceryl caprylate/caprate, 10%), and the

mixture was stirred for 15 min at room temperature. Unl-ME and LEE-ME were prepared, using a concentration of LEE at 1% (*w/w*) related with the total weight of formulation.

Besides the organoleptic properties of ME (aspect, color, transparency), the droplet size and the PdI of the prepared formulations were analyzed through a Zetasizer NanoS (Malvern Instruments, Malvern, UK) at 25.0 ± 0.1 °C. The analysis was conducted without dilution of ME. The refractive index and viscosity of each formulation was previously determined for adjusting the settings of dispersant material required for the analysis of droplet size in the ME. Experiment was performed in triplicate.

### 2.6. Preparation and Characterization of Macroemulsions

For the preparation of macroemulsions, the oily phase was prepared by blending 10% of triglycerides of medium-chain, 10% of phenoxyethyl caprylate and 0.63% of sorbitan esters. The aqueous phase constituted by 2% of thickening-stabilizing-texturizing, 1.5% of O/W sugar-based emulsifier, 0.37% of polysorbate, 4% of glycol, 1.5% of preservative and water until perform a total mass of 100% were also prepared by weighting each compound. After preparation both phases, the oily phase was added to the aqueous phase and manually stirred at room temperature until emulsion formation. The pH of the formulations was adjusted with NaOH 5N until a final pH between 5.0 and 5.5, using a WTW™ SenTix™ 81 pH Electrodes. For the preparation of LEE-Macro, the LEE was added to the oily phase at 1% (*w/w*) related with the total mass of formulation. The aspect and color of macroemulsions were observed. The distribution of the droplet size (d(10), d(50), d(90)) of Unl-Macro and LEE-Macro was assessed through a Malvern Mastersizer 2000 (Malvern Instruments, Malvern, UK) coupled with a Hydro S accessory. For this analysis a default refractive index of 1.52 was selected and the span was posteriorly calculated as described by Carriço et al. (2019) [22]. Additionally, dynamic viscosity measurements were performed through a Kinexus Lab+ Rheometer (Malvern Instruments, Malvern, UK) employing a cone-and-plate geometry. The measurements were performed between 1 and 1000 Pa on a logarithmic increment ranging from 1.0 to 100 s$^{-1}$. Oscillation frequency sweep was also performed using the same geometry and frequencies ranging between 0.1 and 1 Hz, on 10 samples per decade.

### 2.7. Sensorial Analysis of Formulations

Twenty-five healthy untrained testers were assigned to assess different formulations, performing a test of the organoleptic characteristics (odor and color) through a questionnaire. The study protocol was designed by the researchers and the procedures followed were in accordance with the ethical standards of the FFULisboa committee. The questions and the scale are as described: the odor intensity (without odor, slightly perceptible, perceptible, very perceptible and intense odor); with exception of volunteers that answered without odor, the volunteers were questioned about odor perception (pleasant and fresh odor, pleasant and hot odor, unpleasant, very unpleasant and undefined); lastly, the color of all formulations were also assessed (unpleasant, slightly pleasant, indifferent, pleasant and very pleasant). The sensory analysis also enabled to study the acceptability of odor and color of each formulation by the volunteers. This parameter was measured by the likelihood of buying different cosmetic products (face cream, body cream, shampoo and toothpaste) with the odor and color of the respective formulation (would never buy, unlikely, likely, quite likely and purchased the product). At the beginning the volunteers performed the sensorial analysis without knowing any information about the composition of the formulations; after being informed (AI) that LEE-ME and LEE-Macro contained a tomato extract with antioxidant properties, the volunteers answered the same questionnaire about the organoleptic properties and the likelihood of buying different cosmetic products (acceptability) with the odor and color of the respective formulation (being designated in this work as LEE-ME (AI) and LEE-Macro (AI)). Knowing that information, the sensorial analysis of these two formulations was re-assessed. All subjects remained in a room free from strong odors, maintained at 25 ± 1 °C and 40–60% relative humidity, under natural

day light, for 5 min prior to evaluation. The samples in test (at room temperature) were placed in front of each volunteer aligned with the sequential questions of the questionnaire.

*2.8. Identification of Volatile Compounds through GC-MS*

The analysis of volatile compounds was carried out by GC-MS, in two different conditions. (1) Initially the analysis was performed using a GC-MS equipment (QP 2010 Plus, Shimadzu, Kyoto, Japan) coupled with an autosampler AOC-5000 and a Teknokroma Sapiens 5MS column (5%-Phenyl- 95% Dimethyl Polysiloxane column) 30 m × 0.25 mm ID, film thickness 0.25 μm and an AOC-5000 Shimadzu autosampler. Helium was used as a carrier gas, and after the splitless injection, runs were performed in a column flow at 2 mL/min, using a temperature gradient for the separation of sample components: column oven was initially maintained at 40 °C during 5 min, followed by a gradual increase of 5 °C/min until 170 °C and programmed to rise to 230 °C at the rate of 30 °C/min; at the end the temperature was kept for 4 min. (2) The volatile compounds of the samples were also analyzed through a GC-MS QP 2010, Shimadzu coupled to an autosampler AOC-5000 Plus, and a TeknoKroma Sapiens Wax MS, 60 m, 0.25 mm (i.d.), 0.25 μm column. Carrier flow was performed at 4 mL/min, with an injection mode in splitless and detector at 250 °C. The gradient temperature used was the same abovementioned. For both conditions, the ionization energy was 70 eV, a scan range of 29–300 $m/z$, while the detector and ionization temperatures were at 250 °C. Each sample was weighted to a 20 mL-vial (around 5–6 mg of LEE, 60–70 mg of both ME and Macro). Then, the volatile compounds present in the headspace of each sample were extracted using solid phase microextraction (SPME). A fiber of DVB/CAR/PDMS from Supelco, 23 Ga, 50/30 μm, gray fiber assembly was used. Samples were heated for 40 min at 40 °C and were injected at 250 °C.

The data acquisition was made using Shimadzu software, GC-MS solution, version 2.10 when analyzed with 5-MS column, while a different version (4.50 SP1) was used for the analysis performed with a Wax column. The identification of the components was assigned through comparison of the peak detected in the chromatogram obtained by GC–MS analysis, according to experimental conditions, with installed spectra from the libraries (NIST 21, 27, 107, 147 and Wiley 229). Only volatile compounds whose spectra presented a similarity index higher than 80% were considered for the comparison of Kovats retention indices. This retention index is used to confirm the identity of each volatile compound. Briefly, an injection of an alkane standard solution $C_8$–$C_{20}$ was performed using the same chromatographic conditions overmentioned. According with the retention time of each alkane, the retention index of each volatile compound in the sample chromatogram was determined based on the equation used for the calculation of non-isothermal Kovats retention indices. The Cal RI were compared with retention index found in the literature (Lit RI), and when the difference between both RI was higher than 15 units, the compound was assigned as "not identified". Considering these criteria, Tables 1 and 2 present the volatile compounds that were identified.

In order to evaluate the contribution of each peak, they were integrated and the relative percentage of the area was determined, considering the total area of all integrated peaks in the chromatogram.

## 3. Results and Discussion

### 3.1. Proximate Composition of Tomato Waste

Tomato waste consisted of 29% seeds and 71% skins. The water content of 5.3 ± 0.04 wt.%, was relatively low, as set by lyophilization of the original waste supplied.

The Soxhlet extraction of tomato waste with n-hexane obtained an oil yield of 11.9 ± 0.5 g oil/100 g dry material. The lycopene content of the extract was found to be 56.5 mg lycopene/100 g extract). Vági et al. (2007) reported an oil yield ranging between 3 to 15 g oil/100 g dry material for several dried tomato waste with n-hexane as the solvent [23]. The lycopene content of their extracts varied between 3.2 and 503.1 mg

lycopene/100 g extract. The wide range of values obtained was explained by the authors by the different areas of cultivation and harvesting year.

### 3.2. Supercritical $CO_2$ Extraction of LEE from Tomato Waste

Tomato waste was submitted to supercritical carbon dioxide extraction at 500 bar and 60 °C. The SFE kinetics is shown in Figure 3. Two distinct parts can be seen in the extraction curve showing that two different mechanisms are acting in the SFE of LEE from tomato waste. In first part of the curve, the oil that is readily available at the solid surface of tomato waste is extracted by $CO_2$ at a fast and constant rate (shown by the relatively constant slope of the curve) indicative that external mass transfer resistance in the $CO_2$-rich phase and equilibrium solubility are controlling the extraction kinetics. Hereafter, diffusional and internal mass transfer resistances dominate the extraction process. Readily available oil gets exhausted and oil from deeper sections of the solid waste start to be extracted by the solvent at a much lower rate.

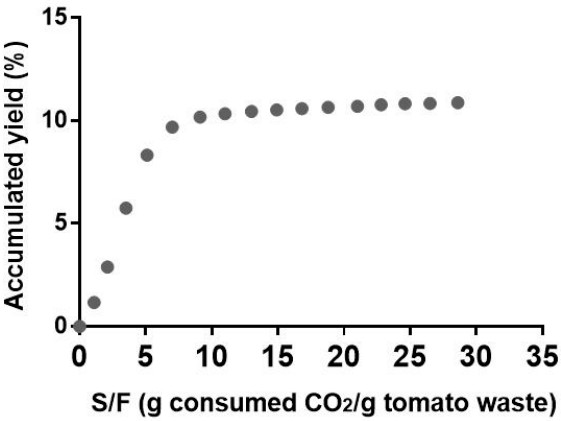

**Figure 3.** SFE extraction kinetics of oil from tomato waste at 500 bar and 60 °C. S/F: mass ratio of solvent to feed.

The overall SFE extraction yield was 10.9 ± 0.4gextract/100g dry waste, accounting for ca. 91% of total oil present in tomato waste, according to Soxhlet extraction with n-hexane.

The fatty acid composition profile of tomato waste oil as determined by GC-FID analysis showed good agreement with the data found in the literature [23–26]. Linoleic acid (57.8 ± 0.5%), oleic acid (23.9 ± 0.3%) and palmitic acid (12.8 ± 0.1%) were the major fatty acids detected. Stearic acid (5.4 ± 0.1%) was present in minor quantity.

Lycopene content of the SFE extract was found to be 50.0 mg lycopene/100 g extract. This corresponds to ca. 81% of recovery of the lycopene presents in tomato waste.

Baysal, Ersus and Starmans (2000) optimized the extraction conditions (pressure, temperature, extraction time and $CO_2$ flow rate) of β-carotene and lycopene from dried tomato paste waste [27]. Higher temperature (65 °C) and higher pressure (300 bar) conditions obtained a maximum lycopene recovery yield of 22%. Rozzi et al. (2002) evaluated the effects of temperature, pressure and $CO_2$ flow rate on the recovery yields of carotenoids from dried tomato by-products (tomato skins and seeds) [26]. They obtained a maximum amount of lycopene extracted of ca. 60% at 86 °C and 345 bar. Vági et al. (2007) obtained similar oil yields either by Soxhlet extraction with n-hexane or supercritical $CO_2$ extraction at 460 bar and 60 °C [23]. A maximum lycopene recovery of ca. 78% was achieved at 460 bar and 100 °C.

### 3.3. Physical Characterization of Formulations

The organoleptic properties of formulations were evaluated. Both microemulsions were liquid and formed spontaneously due to the high amount of surfactant. Unloaded microemulsion (Unl-ME) was clear and transparent, while the inclusion of LEE in the ME originated a clear formulation with a yellowish color (Figure 4A). The droplet size and

polydispersity index (PdI) of both microemulsions were also evaluated: Unl-ME presented a droplet size of 5.5 ± 0.1 nm and a PdI of 0.15 ± 0.01. The incorporation of LEE into the microemulsions enabled the formation of LEE-loaded microemulsions (LEE-ME) with a droplet size and a PdI of 8.4 ± 0.4 nm and 0.45 ± 0.06, respectively. The presence of droplet size smaller than 100 nm is a described characteristic of microemulsions [28]. The incorporation of LEE did not change the ME droplet distribution. On the other hand, macroemulsions displayed a homogeneous semi-solid aspect. Unloaded macroemulsions (Unl-Macro) were white, while LEE-loaded macroemulsions (LEE-Macro) presented a yellowish color due to the presence of LEE (Figure 4A). The presence of LEE did not change the droplet size distribution and it was observed that 90% of the droplets of both macroemulsions displayed a particle size higher than 1.4 um (d(10), Table S1)). Therefore, these formulations were considered coarse systems, since the internal droplet size is higher than 500 nm [28]. It was also observed that Unl-Macro and LEE-Macro presented shear thinning behavior (Figure 4B), since the viscosity decreased with the increase of the shear rate [29], as well as a similar elastic (G') and the viscous (G'') modulus (Figure 4C). Thus, the presence of LEE did not influence the rheological properties of these macroemulsions.

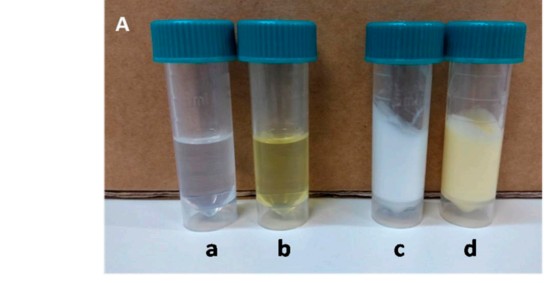

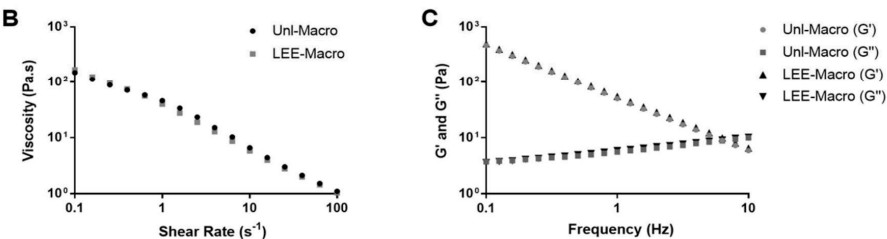

**Figure 4.** (**A**) Photograph images of ME and macroemulsions: Unl-ME (a), LEE-ME (b), Unl-Macro (c) and LEE-Macro (d); (**B**) Effect of shear rate on apparent viscosity of macroemulsions; (**C**) Effect of oscillation frequency sweep test in the G' and G'' in macroemulsions.

### 3.4. Sensorial Analysis of Formulations

The odor and color of formulations were evaluated by 25 untrained volunteers. The distribution of age and sex of the volunteers is described in Figure S1 of Supplementary Material. The acceptability of the formulations was assessed by measuring the likelihood of buying different personal care products, namely face cream, body cream, shampoo and toothpaste, with the odor and the color of respective formulation (Tables S2 and S3 of Supplementary Material).

### 3.4.1. Odor Analysis of Formulations

The organoleptic properties were evaluated by untrained volunteers. In this preliminary study, the odor of Unl-ME was mainly classified as without odor. However, the odor of LEE-ME was classified as a slightly perceptible and perceptible odor, indicating that LEE confers odor to the ME. LEE-ME odor after volunteers being informed about formulation composition (AI, see Section 2) presented a pattern similar in odor intensity comparatively to LEE-ME before the information (Figure 5A), indicating that the information about the composition of LEE-ME did not change the odor perception of ME.

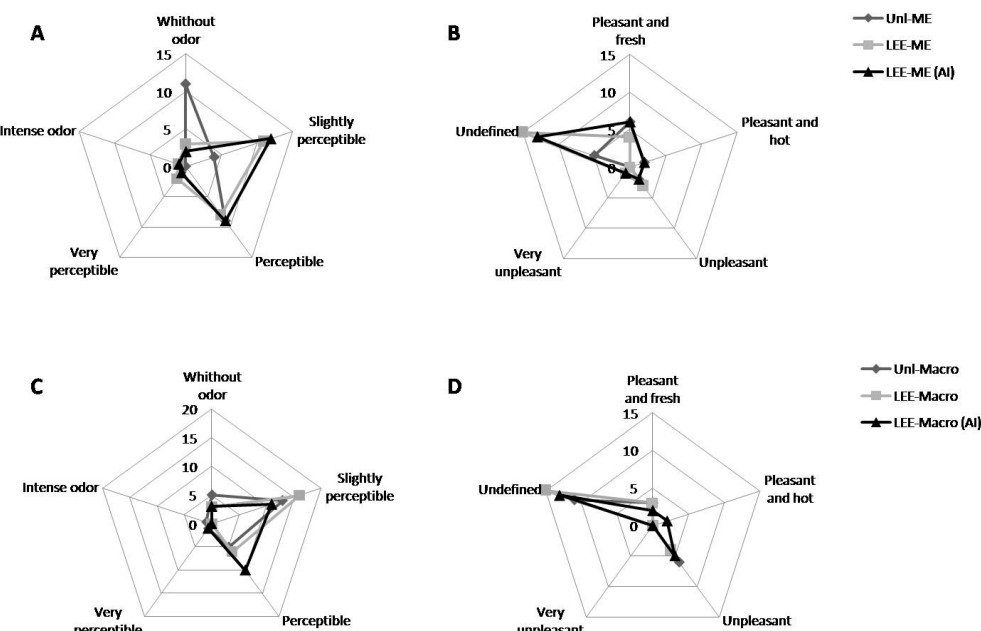

**Figure 5.** Sensorial analysis of odor in ME (**A**,**B**) and macroemulsions (**C**,**D**). The odor intensity (**A**,**C**) and the classification of odor (**B**,**D**) of formulations were evaluated by the 25 untrained volunteers.

The volunteers that identified odor in the ME were questioned about the type of odor (Figure 5B). There was no consensus about the odor of Unl-ME since all classification was observed. The presence of LEE influenced the odor perception when compared with Unl-ME: initially the majority classified the odor as undefined, and after being informed about the LEE-ME composition (LEE-ME (AI)), the majority of volunteers still answered as undefined odor. The odor perception was quite similar before and after information, being classified as a pleasant and fresh/hot, unpleasant or even very unpleasant (Figure 5B). Therefore, the presence of LEE into the ME changed the odor intensity and its perception, but the information about the composition of LEE-ME did not modify the odor perception.

The likelihood of buying different personal care products with the odor of each ME was also evaluated in the same 25 untrained volunteers (Table S2 of supplementary material). The likelihood of buying a face and body cream with the odor of LEE-ME was the same or even lower when compared with Unl-ME, respectively. However, after information about the composition of LEE-ME, it was possible to observe an increase in the percentage of answers "likely" and "quite likely". The same profile was also observed for likelihood of buying a shampoo: the percentage of answers "would never buy" was higher for LEE-ME than the respective Unl-ME, but the percentage of answers "likely" and "quite likely" increased for LEE-ME (AI). Concerning the toothpaste, overall, the likelihood of buying a product with the odor of ME was low, since around 80% of volunteers either answered "would never buy" or "quite likely". The information about the composition of LEE-ME presented a positive impact in the acceptability of some cosmetic/personal care product, but not in all products, such as toothpaste, where the undefined odor of formulations can be the main limitation.

The odor intensity of Unl-Macro and LEE-Macro was mainly classified as slightly perceptible. When informed about the composition of LEE-Macro (LEE-Macro (AI)) the odor intensity slightly changed, being mainly classified either as a slightly perceptible or as perceptible (Figure 5C). Concerning the odor perception, the majority identified the odor as undefined for all macroemulsions and a small number of volunteers identified the odor as unpleasant, pleasant and fresh or even pleasant and hot. Contrary to the ME, the presence of LEE did not influence the odor perception in the LEE-Macro, neither in LEE-Macro (AI) (Figure 5D).

The likelihood of buying a face cream, a body cream or a shampoo with the odor of the LEE-Macro was higher than Unl-Macro, since LEE-Macro showed higher percentage of answers "likely". Moreover, the answers "quite likely" and "would buy" increased for LEE-Macro (AI) when compared to Unl-Macro and LEE-Macro, indicating that the information about the composition of LEE-Macro influenced the acceptability of these products. Concerning the toothpaste, the sum of answers "would never buy" and "unlikely" was more than 75% for the Unl-Macro, LEE-Macro and LEE-Macro (AI) (Table S2 of Supplementary Material), suggesting that the odor of macroemulsions with and without LEE was not suitable for the development of this personal care product.

The odor of LEE-ME and LEE-Macro could potentially be used for the development of face and body creams, or even shampoos. The information about formulation composition, namely the presence of a natural antioxidant might influence the acceptability of a perceptible odor. In the case of toothpastes, this information may not have any impact, where traditional flavors such as mint, peppermint or eucalyptus are preferred by general consumers [30]. The presence of flavor is also responsible for the feeling of breath-freshening, that allows to mask mouth odors or even the taste of some ingredients present in the formulation [31].

### 3.4.2. Color Analysis of Formulations

The majority of the volunteers answered that the color of Unl-ME was indifferent or pleasant, while the incorporation of LEE into the ME originated an equal distribution in the answers. Many volunteers referred that the liquid aspect and the yellowish color of LEE-ME originated an unpleasant formulation, although the physical aspect of formulation was not the focus of this work. However, after being informed about the composition of LEE-ME (LEE-ME (AI)), there was a reduction of the answers unpleasant and slightly unpleasant, and an increase in the number of answers indifferent, pleasant and very pleasant (Figure 6A). Regarding the likelihood of buying a body cream, a face cream and a shampoo with the color of ME, it is possible to observed that Unl-ME presented higher percentage of answers "quite likely" and low percentage of answers "would never buy" than LEE-ME (Table S3 of Supplementary Material). Nevertheless, the percentage of answers "quite likely" increased, while the percentage of answers "would never buy" and "quite unlikely" reduced for LEE-ME (AI) comparatively to LEE-ME. On the other hand, the likelihood of buying a toothpaste was lower for all ME (the sum of answers "would never buy" and "unlikely" totalize more than 80%). This study suggests that the color conferred by the LEE originated a broad classification (from unpleasant to pleasant), but the information about the composition of LEE-ME changed the classification in a positive way, as well as the likelihood of buying some cosmetic products with the color of LEE-ME. The color in all macroemulsions was classified as pleasant or indifferent. Three volunteers answered that the color of LEE-Macro was slightly unpleasant, but after information about the composition of LEE-Macro (LEE-Macro (AI)), the number of answers very pleasant increased and none of the volunteer answered slightly unpleasant (Figure 6B). The acceptability of the color of macroemulsions was also evaluated by measuring the likelihood of buying different cosmetic and personal care products. The likelihood of buying face and body creams with the color of Unl-Macro was higher than the LEE-Macro: the percentage of answers "quite likely" and "would buy" were almost the double for Unl-Macro (Table S3 of supplementary material). The presence of color also reduced the likelihood of buying a shampoo: the percentage of answers "would buy" had a reduction of 16% in the case of LEE-Macro when compared to Unl-Macro. Nevertheless, in the case of LEE-Macro (AI) it was possible to observe an increase in the answers "would buy" for the face cream, body cream and shampoo. Regarding the likelihood of buying toothpaste with the color of macroemulsions, Unl-Macro presented a high likelihood than LEE-Macro and LEE-Macro (AI), indicating that the yellow color of macroemulsions negatively influenced the acceptability of toothpastes, even when the color had been classified as pleasant. Although macroemulsions either with a white or yellowish color presented a pleasant color,

there is a higher likelihood of commercializing white cosmetic products. The information about the composition of the LEE-Macro and the benefic skin effect can be useful for the product acceptability by the consumers.

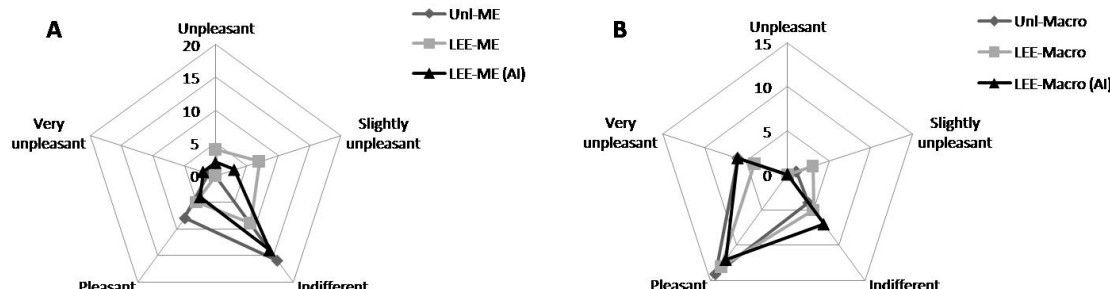

**Figure 6.** Sensorial analysis of the color in ME (**A**) and macroemulsions (**B**) evaluated by the 25 untrained volunteers.

### 3.5. Identification of Volatile Compounds in the Formulations

The flavor and odor of tomato is widely dependent by the tomato varieties, the cultivars conditions, the stage of ripeness, the harvesting, the processing and storage [19,21]. Tomatoes contain several volatile compounds responsible for the aroma [19,32], but the processing of this feedstock may change the composition of aroma volatile compounds, and consequently the organoleptic properties. In this work the tomato waste was initially lyophilized, and the LEE was extracted through SFE at 60 °C/500 bar, for being later incorporated into different formulations. The crop conditions of tomato, harvesting, storage as well as the extraction method, which use temperature and may promote the evaporation of some volatile compounds or the formation of new compounds [33] are some explanation by the absence or presence of some volatile compounds in the LEE, which has never been described in the literature. Marković et al. (2007) demonstrated that the concentration of (Z)-3-Hexenal, one of main aroma-active compounds responsible for the aroma of fresh tomato, decreased when the tomato is processed, while the amount of furfural increased. Besides that, volatile compounds such as geranial, benzyl alcohol and 6-metyl-5-hepten-2-one have been found in tomato processing products (juice, paste, puree and canned diced tomatoes), but in less amount than in fresh tomatoes [34]. Kelebek et al. (2018) also demonstrated that during the tomato processing some compounds may be formed and some volatile compounds could be lost by evaporation [33]. Some of the volatile compounds expected for the tomato extract were not found, maybe because their concentration was below the limit of detection when analyzed by gas chromatography–mass spectrometry (GC-MS).

The identification of volatile compounds in the LEE and formulations (microemulsions and macroemulsions) was performed through SPME-GC-MS analysis, using two columns with different polarities, one with an intermediate polarity (5-MS column) and another polar (Wax column), enabling different interactions of volatile compounds with both stationary phases and a greater coverage in the detected number of compounds. Some important volatile compounds, usually described in the literature for fresh tomato and its processing products, were not identified in LEE analysis, namely, (Z)-3-Hexenal, (E)-2-hexenal, 3-methylbutanal, 1-octen-3-one, methional and 1-penten-3-one [19,35]. On the other hand, the GC-MS analysis allowed the identification of volatile compounds in the free LEE (1-nitropentane, 2,3-butanediol, 1-pentanol, (Z)-2-penten-1-ol, phenylacetaldehyde, methyl salicylate, furfuryl alcohol and 4-methyl-5H-furan-2-one; data not shown) but not in the ME and macroemulsions formulations. Since this work aims to identify the volatile compounds of free LEE that are responsible for the smell of LEE-ME and LEE-macroemulsions, only the volatile compounds that were simultaneously identified in the LEE-ME and LEE-Macro will be mentioned. The compound 2-phenoxyethanol was identified in small amounts in LEE; however, it presented the highest relative area in the LEE-Macro and Unl-Macro formulations. The identification of this compound is related with the presence of phenoxyethyl caprylate, an ester of 2-phenoxyethanol used

as an emollient for the preparation of the emulsion oily phase. The library used for the identification of volatile substance in the GC-MS analysis did not contain phenoxyethyl caprylate, and therefore, it only allowed the identification of compounds with similar structures, in this case, the 2-phenoxyethanol.

The identified volatile compounds responsible for the odor of Unl-ME and LEE-ME are described in Table 1. Through this analysis it was possible to identify dodecane (with the highest relative area) and D-limonene, either in LEE-ME and Unl-ME. However, some of specific volatile compounds were only identified in LEE-ME: 6-Methyl-5-hepten-2-one (with the higher relative area), 2-isobutylthiazole, (E)-citral (with a low relative area) and citronellyl formate. Other volatile compounds (2,5-dimethyldecane and 1,7-dimethyl-4-(1-methylethyl) cyclodecane) identified in LEE-ME have not been described in the literature. The type of odor/description has not been found, and since they are present in relative low amount (low relative area), it is possible that they may not have an impact in the odor perception of ME.

The absence of consensus regarding the odor of Unl-LEE may be attributed by the combination of different odors, in particular by the combination of dodecane [36], that conferred an alkane-like odor (undefined and unpleasant odor) and D-limonene that presents a much pleasant citric odor [36].

The undefined odor of LEE-ME identified by the majority of the volunteers may be explained by the combination of different type of odors: citric, lemon, floral and fermented/musty odors. Based on the data from literature, the most relevant compounds responsible for the lemon, citrus and floral odors of LEE-ME are 6-methyl-5-hepten-2-one, D-limonene, (E)-citral [35,36] and citronellyl formate [37], which could be the responsibility by the most pleasant and fresh odors. LEE-ME also contained 2-isobutylthiazole, a relevant compound that contributes for the tomato aroma, which is the responsible for the characteristic smell of tomato's leafy and fermented odor [19,36]. Although the relative area of 2-isobutylthiazole is lower than the other volatile compounds with a citric/floral odor, its odor threshold is much lower, indicating that it could be better detected in lower amounts in comparison with the D-limonene or 6-methyl-5-hepten-2-one. 2-Isobutylthiazole is sulfur-derived volatile compound and is responsible for the flavor of horseradish type and spoiled vine-like [38], being one the main reasons for the unpleasant odor of the LEE-ME, initially identified by the volunteers (three volunteers classified the odor as unpleasant and one of them as very unpleasant). The relative high amounts of dodecane in the LEE-ME may also contribute for the undefined or even unpleasant odor of LEE-ME.

Regarding the analysis of volatile compounds in Unl-Macro, it was possible to find different types of compounds with different odor types (Table 2): volatile compounds with a citric and floral odor, namely D-limonene [36], benzyl alcohol [33,34] and phenylethyl alcohol [30] that may be responsible for pleasant and fresh odor identified by three volunteers; volatile compounds, such as dodecane [36], n-hexanal [19,36], 1-hexanol [33] and (E)-2-Octenal [33], that provided an unpleasant odor identified by six volunteers. Although n-hexanal and (E)-2-Octenal are in relative low amounts in Unl-Macro, the odor of these volatile compounds could be easily percept by the volunteers because they present a low odor threshold when compared with the odor compounds with a better pleasant odor (D-limonene, benzyl alcohol and phenylethyl alcohol). Therefore, the combination of these different types of odors may be the explanation why the majority of the volunteers classified the odor as undefined.

In the LEE-Macro it was possible to identify several compounds that are well described in the literature: 6-Methyl-5-hepten-2-one [35], D-limonene [36], 2-isobutylthiazole [19,36], benzeneacetaldehyde [20], dodecane [36], citral [35], toluene [39], n-hexanal [19,36], 1-hexanol [33], (Z)-3-hexanol [33,35], (E)-2-octenal [33] and benzyl alcohol [33,34]. Each compound presents a different odor type, from pleasant odors (citric and floral) to more unpleasant odors (green, musty and fatty) (Table 2). Citronellyl formate and phenylethyl alcohol have been identified in the LEE and LEE-Macro, but they have not been described for tomato and tomato-derived products. Other volatile compounds were found only in

the LEE and LEE-Macro in relative low amounts (2-propyl-1-pentanol, 2,5-dimethyldecane, 3-(3,3-dimethylbutyl)cyclohexanone, 2,4,4,6,6,8,8-hHeptamethyl-2-nonene and 2-bromo dodecane). However, these compounds were not identified in the tomato fruits and tomato-processing products, and the odor type/description has not been described in the literature.

The presence of different volatile compounds in the LEE-Macro with different odor types is the main reason for the undefined odor described by 15 of the volunteers. Nevertheless, three of the volunteers classified the odor of LEE-Macro as pleasant and fresh, probably due to the presence of compounds such as 6-methyl-5-hepten-2-one, D-limonene and citral, while the unpleasant odor described by four volunteers was probably caused by compounds with a musty and fatty odor (2-isobutylthiazole, n-hexanal and (E)-2-Octenal). These three mentioned compounds were not the compounds with the highest relative area but are the volatile compounds with a high odor threshold, being perceived by the volunteers in lower amount when compared with the compounds with a much pleasant odor, higher relative area and odor threshold (6-Methyl-5-hepten-2-one and D-limonene).

The degradation of carotenogenic materials (carotenoids and lycopene) through heat treatments originates volatile compounds that are responsible by the aroma, such as 6-methyl-5-hepten-2-one, β-ionone and citral [40,41]. Citral-derived compounds, such as citronellol have been identified in oleoresins of *Lycopersicum esculemtum* [41] and ROMA VF varieties [39]; however, it was not identified in the GC-MS analysis in this study. Citronellyl formate is another citral-derived compounds that can be prepared by esterification of citronellol with formic acid [42], and its aroma is described a fruity, floral and rose-light odor [43]. Citronellyl formate has not been described in the fresh tomato and their processing products in the literature, but it was identified either in free LEE, LEE-ME and LEE-Macro. Its presence in the LEE may be due to the oxidative degradation of carotenoids and lycopene caused by the temperature and pressure during the extraction process, or eventually by the enzymes during the harvesting, storage or extraction of the tomato waste [44].

**Table 1.** Characterization parameters of volatile compounds in LEE, LEE-ME and Unl-ME identified through GC-MS analysis.

| | Compound | Area of the Peak (%) | | | Cal RI | Lit RI [1] | Odor Type/Description | Odor Threshold (ppb) [2] | Odor Threshold in Tomato Fruits (ppb) [3] | Reference |
| | | LEE | LEE-ME | Unl-ME | | | | | | |
|---|---|---|---|---|---|---|---|---|---|---|
| 5-MS Column | 6-Methyl-5-hepten-2-one | 34.19 | 5.6 | - | 990 | 986 | Floral, green | 50 | 50 | [35] |
| | D-Limonene | 2.9 | 1.64 | 0.92 | 1028 | 1028 | Lemon, Orange | 10 | 10 | [36] |
| | 2-Isobutylthiazole | 3.14 | 0.66 | —— | 1033 | 1020 | Tomato leafy, green, musty | 2–3.5 | 3.5 | [19,36] |
| | 2,5-Dimethyldecane | 2.11 | 1.78 | —— | 1088 | 1086 | —— | —— | —— | —— |
| | Dodecane | 0.2 | 17.06 | 25.76 | 1200 | 1200 | Alkane | —— | —— | [36] |
| | Citronellyl formate | 1.12 | 0.24 | —— | 1276 | 1275 | Fruity, floral, rose-light odor | —— | —— | —— |
| | Not identified | 2.14 | 2.08 | —— | —— | —— | —— | —— | —— | —— |
| | Not identified | 0.15 | 0.09 | —— | —— | —— | —— | —— | —— | —— |
| | Not identified | 0.12 | 0.27 | —— | —— | —— | —— | —— | —— | —— |
| | Not identified | 1.88 | 2.17 | —— | —— | —— | —— | —— | —— | —— |
| | Not identified | 0.37 | 0.47 | —— | —— | —— | —— | —— | —— | —— |
| | Not identified | 0.25 | 0.26 | —— | —— | —— | —— | —— | —— | —— |
| | Not identified | 0.37 | 0.31 | —— | —— | —— | —— | —— | —— | —— |
| | Not identified | 0.26 | 0.36 | —— | —— | —— | —— | —— | —— | —— |
| | 1,7-Dimethyl-4-(1-methylethyl)cyclodecane | 0.17 | 0,13 | —— | 1478 | 1485 | —— | —— | —— | —— |
| | Not identified | 0.33 | 0.3 | —— | —— | —— | —— | —— | —— | —— |
| | Not identified | 0.04 | 0.5 | —— | —— | —— | —— | —— | —— | —— |
| Wax Column | D-Limonene | 1.27 | 0.21 | 0.06 | 1202 | 1198 | Lemon, Orange | 10 | 10 | [36] |
| | Not identified | 2.39 | 0.3 | —— | —— | —— | —— | —— | —— | —— |
| | 2-Isobutylthiazole | 4.9 | 0.20 | —— | 1410 | 1406 | Tomato leafy, green, musty | 2–3.5 | 3.5 | [19,36] |
| | Not identified | 1.2 | 0.13 | —— | —— | —— | —— | —— | —— | —— |
| | €-Citral | 2.02 | 0.06 | —— | 1740 | 1741 | Citrus, Lemon | 32 | —— | [35] |

Area of the peak expressed as relative percentage of the total area of all integrated peaks in the chromatogram; Cal RI—Calculated Retention Index; Lit RI—Retention Index from literature; [1] Lit RI from Pubchem and ChemSpider databases [40,41]; [2] Odor threshold based on the Leffingwell & Associates website [39]; [3] Odor Threshold in water of volatile compounds identified in tomato fruits [36].

**Table 2.** Characterization parameters of volatile compounds in LEE, LEE-Macro and Unl-Macro identified through GC-MS analysis.

| | Compound | Area of the Peak (%) | | | Cal RI | Lit RI [1] | Odor Type/Description | Odor Threshold (ppb) [2] | Odor Threshold in Tomato Fruits (ppb) [3] | Reference |
| | | LEE | LEE-Macro | Unl-Macro | | | | | | |
|---|---|---|---|---|---|---|---|---|---|---|
| **5-MS Column** | Not identified | 0.14 | 0.08 | 0.05 | — | — | — | — | — | — |
| | 6-Methyl-5-hepten-2-one | 33.95 | 1.59 | — | 990 | 986 | Floral, green | 50 | 50 | [35] |
| | D-Limonene | 2.9 | 1.86 | 1.27 | 1028 | 1028 | Lemon, Orange | 10 | 10 | [36] |
| | 2-Isobutylthiazole | 3.14 | 0.1 | — | 1033 | 1020 | Tomato leafy, green, musty | 2–3.5 | 3.5 | [19,36] |
| | Benzeneacetaldehyde | 2.44 | 0.09 | — | 1043 | 1035 | Green | 4 | — | [20] |
| | 2-Propyl-1-pentanol | 0.14 | 0.06 | — | 1059 | 1052 | — | — | — | — |
| | 2,5-Dimethyldecane | 2.11 | 0.05 | — | 1088 | 1086 | — | — | — | — |
| | Dodecane | 0.2 | 0.33 | 0.7 | 1200 | 1200 | Alkane | — | — | [36] |
| | 2-Phenoxyethanol * | 0.22 | 14.59 | 23.64 | 1226 | 1226 | — | — | — | — |
| | Citral | 1.2 | 0.17 | — | 1271 | 1270 | Citrus, Lemon | 30–32 | — | [35] |
| | Citronellyl formate | 1.12 | 0.12 | — | 1276 | 1275 | Floral | — | — | — |
| | Not identified | 2.14 | 0.16 | — | — | — | — | — | — | — |
| | 3-(3,3-Dimethylbutyl)cyclohexanone | 1.38 | 0.04 | — | 1352 | 1364 | — | — | — | — |
| | 2,4,4,6,6,8,8-Heptamethyl-2-nonene | 1.88 | 0.08 | — | 1362 | 1343 | — | — | — | — |
| | 2-Bromo dodecane | 0.33 | 0.02 | — | 1522 | 1505 | — | — | — | — |
| **Wax Column** | Toluene | 3.43 | 1.05 | — | 1047 | 1044 | Sweet | — | — | [39] |
| | n-Hexanal | 1.31 | 0.20 | 0.91 | 1090 | 1091 | Green, grassy | 4.5–5 | 4.5–5 | [19,36] |
| | D-Limonene | 1.27 | 1.42 | 0.47 | 1202 | 1198 | Lemon, Orange | 10 | 10 | [36] |
| | Not identified | 2.39 | 0.15 | — | — | — | — | — | — | — |
| | 1-Hexanol | 0.9 | 0.42 | 0.26 | 1358 | 1358 | Herbal | 1500 | — | [33] |
| | (Z)-3-Hexenol | 0.77 | 0.11 | — | 1388 | 1388 | Green, Cut grass | 70 | — | [33,35] |
| | 2-Isobutylthiazole | 4.9 | 0.38 | — | 1410 | 1406 | Tomato leafy, green, musty | 2–3.5 | 3.5 | [19,36] |
| | (E)-2-Octenal | 0.66 | 0.4 | 0.3 | 1437 | 1437 | Fatty | 3 | — | [33] |
| | Citronellylformate | 1.86 | 0.15 | — | 1629 | 1629 | Floral | — | — | — |
| | (E)-Citral | 2.02 | 0.18 | — | 1740 | 1741 | Citrus, Lemon | 32 | — | [35] |
| | Benzyl Alcohol | 0.25 | 0.08 | 0.06 | 1885 | 1882 | Floral | 10000 | — | [33,34] |
| | Phenylethyl Alcohol | 0.69 | 0.15 | 0.03 | 1917 | 1917 | Floral | 750–1100 | — | — |

Area of the peak expressed as relative percentage of the total area of all integrated peaks in the chromatogram; Cal RI—Calculated Retention Index; Lit RI—Retention Index from literature; [1] Lit RI from Pubchem and ChemSpider databases [40,41]; [2] Odor threshold based on the Leffingwell & Associates website [39]; [3] Odor Threshold in water of volatile compounds identified in tomato fruits [36]. * 2-Phenoxyethanol was identified since it presents a chemical structure similar to phenoxyethyl caprylate used for the preparation of macroemulsions. Phenoxyethyl caprylate it is not present in the library and therefore the software identified a similar compound.

## 4. Conclusions

The cosmetics industry has an increasing interest in the incorporation of natural complex mixtures such as plant extracts, which may impact the color and odor of the final products.

In this work, LEE were extracted from tomato waste and incorporated in preparations for topical application. Supercritical $CO_2$ extraction of tomato waste was carried out at 500 bar and 60 °C with an overall extraction yield of ca. 91% of total oil present in tomato waste. The major fatty acids detected in the extracted oil were linoleic acid (57.8%), oleic acid (23.9%) and palmitic acid (12.8%). The lycopene content of the SFE extract was found to be 50.0 mg lycopene/100 g extract, which corresponds to ca. 81% of the lycopene present in tomato waste.

The presence of LEE enables the modification of the organoleptic properties of both formulations, especially the odor and color, but it did not affect the physical properties of formulations. In this work, we report for the first time a study on the organoleptic characteristics of LEE to be incorporated in cosmetic products correlated with the presence of volatile compounds. The odor intensity and perception of ME changed when LEE was incorporated in the oily phase, but it did not affect the odor of LEE-Macro when compared with Unl-Macro. The majority of the volunteers classified the odor of LEE-ME and LEE-Macro as undefined, which could be attributed to the mixture of volatile compounds with pleasant citric and floral odor (D-limonene, citral, 6-methyl-5-hepten-2-one and citronellyl formate) and compounds with unpleasant green, musty and fatty odors (n-hexanal, (Z)-3-hexanol and 2-isobutylthiazole). After informing the volunteers that LEE-ME and LEE-Macro contained a tomato extract with antioxidant properties, the odor intensity and perception did not modify, but increased the acceptability of some cosmetic products (face cream, body cream and shampoo) with the odor of the formulations tested. The colors of the formulations were affected by the presence of LEE. The yellow color of LEE-ME, associated with the liquid aspect of this formulation, divided the opinion of the volunteers. On the other hand, the yellowish color of LEE-Macro was considered pleasant, similar to Unl-Macro, but still the volunteers showed preferences in buying cosmetic products with the white color. The acceptability of the products with the yellow color was influenced when the volunteers were informed about the composition and health properties of these formulations. Additionally, in this preliminary acceptability study it was possible to observe that the acceptability for the same color and odor is strongly dependent on the type of cosmetic product.

**Supplementary Materials:** The following are available online at https://www.mdpi.com/article/10.3390/app11115120/s1, Figure S1: Distribution of the age (A) and sex (B) of untrained volunteers in the sensorial analysis (*n* = 25), Table S1: Cumulative droplet size distribution of macroemulsions, Table S2: Percentage of the three most frequent responses regarding the likelihood of buying different personal care products with the odor of formulations, Table S3: Percentage of the three most frequent responses regarding the likelihood of buying different personal care products with the color of formulations.

**Author Contributions:** Conceptualization, S.S. and H.M.R.; methodology, A.C., F.C., S.S., M.M., J.M., A.P., P.S., M.R.B. and A.F.; software, A.C. and A.F.; validation, A.C., A.F. and J.M.; formal analysis, A.C.; writing—original draft preparation, A.C., M.M. and S.S.; writing—review and editing, P.S., A.P., H.M.R. and J.M.; supervision, S.S. and H.M.R.; project administration, S.S. and H.M.R.; funding acquisition, S.S. and H.M.R. All authors have read and agreed to the published version of the manuscript.

**Funding:** This research was funded by Fundação para a Ciência e a Tecnologia through UIDB/QUI/50006/2020, UIDB/04138/2020, UIDP/04138/2020, PTDC/SAU-SER/30197/2017LISBOA-01-0145-FEDER-030197 and IF/01146/2015.

**Institutional Review Board Statement:** This protocol respected the Helsinki Declaration and Good Clinical Practice studies on products for sensory studies. The study protocol (ner. 4/2021) was submitted to the Ethical Committee from the FFULisboa, however, ethical review and approval were

waived for this study, since the volunteers did not apply any substance or product to any part of their body and the test was completely non-invasive. The tested formulations are free of toxic products.

**Informed Consent Statement:** Informed consent was obtained from all subjects involved in the study.

**Data Availability Statement:** Not applicable.

**Acknowledgments:** The authors are grateful to Italagro S.A. and Marta Bento for providing the processing-industry tomato waste.

**Conflicts of Interest:** The authors declare no conflict of interest. Funders had no role in the design of the study; in the collection, analyses, or interpretation of data; in the writing of the manuscript, or in the decision to publish the results.

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
