# Peer review of "Evaluating the Presence of Lycopene-Enriched Extracts from Tomato on Topical Emulsions: Physico-Chemical Characterization and Sensory Analysis"

_applsci, doi:10.3390/app11115120_

Round 1
Reviewer 1 Report
Introduction: A good rationale has been presented overall. Remove Table 1, as showing the names of a few products (with unknown quantities of lycopene) sold at the moment in one country is not a reasonable rationale for such investigation and is irrelevant to the global readership of this journal. However, some peer reviewed studies on the use/efficacy of lycopene in cosmetics will be appropriate to justify the development of the material in the first place. The introduction section requires thorough proofreading for grammar and sentence structure.
Methods: described with a great level of detail (apparatus settings). Overall, other than the sentence over lines 155-157. The term “probability” has been used in the methods, abstract and results to refer to asking the sensory assessors to report the “likelihood” of buying a product. I assume this is a matter of translating adequately the question from Portuguese to English. I suggest to use “likelihood” as it describes intentions, whilst probability is more commonly used in terms of statistical probability of the occurrence of an event. Check sentence in line 216 as its meaning is unclear.
Results: very detailed overall, often referring to publications of relevance. However, the references and results of this study could be more explicitly directly compared.
The text needs proofreading for minor grammar errors. 380-388 – this paragraph refers to using lycopene and anti-oxidants more broadly in shampoos and toothpastes. There is no scientific justification in the introduction for the benefits of such inclusion. What will benefit from lycopene: the teeth, the hair fibre, the scalp? Hence, reporting purchasing intentions makes no sense, as the consumers are even less likely to know of any possible benefit. The sensory analysis is quite worldly. I advise to report % of combined results which can be grouped as desirable vs % combined undesirable results for each radar-gram, instead of referring to the exact number of people who selected each option or another.
It would help to explain and offer a hypothesis about the origin of the VOCs of the unloaded emulsions – is there a possible contamination?
Conclusion
Clear and well written, the English is better than at the Intro section.

Author Response
- Introduction: A good rationale has been presented overall. Remove Table 1, as showing the names of a few products (with unknown quantities of lycopene) sold at the moment in one country is not a reasonable rationale for such investigation and is irrelevant to the global readership of this journal. However, some peer reviewed studies on the use/efficacy of lycopene in cosmetics will be appropriate to justify the development of the material in the first place. The introduction section requires thorough proofreading for grammar and sentence structure.
Answer: We appreciate your time in reviewing this manuscript. Table 1 was removed as requested. Some studies regarding the use of lycopene in cosmetics were inserted in the introduction section and the text is highlighted in yellow. A proofreading was made to improve introduction´s grammar and sentence structure.
Before modifications:
There are several marketed products claiming the tomato based-origin and the tomato beneficial properties and tomato scent does not appear as an acceptability burden. Table 1 presents commercial products containing tomato-derived ingredients.
After modifications:
There are several marketed products claiming the tomato based-origin and the tomato beneficial properties [8-13] and tomato scent does not appear as being an acceptability burden for these products. In fact, to the best of our knowledge a tomato scent acceptability study in cosmetics has never been reported. Several studies have based their aim on the efficacy of lycopene, mostly combined with other bioactives, on skin ageing or skin photoprotection. The oral consumption of a mixture of soy isoflavones, lycopene, vitamin C, vitamin E and fish oil, clinically demonstrated to improve the depth of facial wrinkles and to increase the deposition of new collagen fibres in the dermis, following long-term use [14]. The possibility of using plant derived bioactives in sunscreens to protect the skin against sun light exposure damage has been hypothesized [15], however, the number of studies including molecules like lycopene in topical forms is very scarce [1]. Apart from lycopene extracted from fruits or food wastes, presenting antioxidant activity, tomato cultured stem cells were developed as a new cosmetic active ingredient and tested for the protection of skin cells towards heavy metal toxicity [16].
- Methods: described with a great level of detail (apparatus settings). Overall, other than the sentence over lines 155-157.
Answer: Thanks for your comments. It is needed to indicate the refractive index and viscosity of dispersant for measuring the droplet size. Most of the time in the DLS (Zetasizer NanoS) analysis, the dispersant material is water, but since the samples were not diluted, we had to calculate such parameters to adjust the parameters. Some modifications in the manuscript have been made related with this information to be clearer.
Before modifications:
The analysis was conducted without dilution of ME, and therefore, the refractive index and viscosity of each formulation was previously determined for adjusting the settings of dispersant material in the Standard Operating Procedures.
After modifications:
The analysis was conducted without dilution of ME. The refractive index and viscosity of each formulation was previously determined for adjusting the settings of dispersant material required for the analysis of droplet size in the ME.
- The term “probability” has been used in the methods, abstract and results to refer to asking the sensory assessors to report the “likelihood” of buying a product. I assume this is a matter of translating adequately the question from Portuguese to English. I suggest to use “likelihood” as it describes intentions, whilst probability is more commonly used in terms of statistical probability of the occurrence of an event. Check sentence in line 216 as its meaning is unclear.
Answer: Regarding the term “probability”, the reviewer has a point. “Likelihood” was used instead.
- Check sentence in line 216 as its meaning
Answer: In the line 216, the sentence was checked and corrected
- Results: very detailed overall, often referring to publications of relevance. However, the references and results of this study could be more explicitly directly compared.
Answer: The discussion was improved in order to better correlate the findings and the references. Several modifications were performed throughout the text and are highlighted in yellow colour.
- The text needs proofreading for minor grammar errors. 380-388 – this paragraph refers to using lycopene and anti-oxidants more broadly in shampoos and toothpastes.
Answer: Thanks for this comment. Modifications in this section were performed.
Before modifications:
Although the information about the LEE in both formulations as well as its antioxidant properties for the skin may not influence the odor perception, it increased the acceptability of these products by the consumers, and consequently will affect their commercialization. Nevertheless, the odor identified at LEE-ME and LEE-Macro is not suitable for the development of toothpastes, where the flavor is an important factor in the commercialization of such products. Different flavors are often used for the formulations of toothpastes, such as mint, peppermint or eucalyptus [27]. The development of toothpastes with flavor is an important factor for its commercialization, since it has a direct impact in the taste and odor which will reflect its acceptability during and after use.
After modifications:
The information about formulation composition, namely the presence of a natural antioxidant might influence the acceptability of a perceptible odor. In the case of toothpastes, this information may not have any impact, where traditional flavours such as mint, peppermint or eucalyptus are preferred by general consumers [30].
- There is no scientific justification in the introduction for the benefits of such inclusion. What will benefit from lycopene: the teeth, the hair fibre, the scalp? Hence, reporting purchasing intentions makes no sense, as the consumers are even less likely to know of any possible benefit.
Answer: We appreciate your observation. As pointed out by the reviewer, there is no scientific justification in the introduction for the benefits of including lycopene in products, such as shampoos and toothpastes nor on the benefits to teeth, hair fibre and scalp. The reason for this is based on the fact that lycopene skin supplementation through topical application is quite new topic. What is possible to find in the literature is the benefits for skin from lycopene oral intake, through food or food supplements. Thus, the information about formulation composition, namely the presence of a natural antioxidant like lycopene, might influence the acceptability of a perceptible odor. In the case of toothpastes, this information may not have any impact, where traditional flavours such as mint, peppermint or eucalyptus are preferred by general consumers.
- The sensory analysis is quite worldly. I advise to report % of combined results which can be grouped as desirable vs % combined undesirable results for each radar-gram, instead of referring to the exact number of people who selected each option or another.
Answer: Thanks for your comment. The results description of sensory analysis in the section 3.4 was shortened and revised. All the modifications are displayed with yellow color. The way of reporting data using grouped of “desirable vs % combined undesirable results” for each graph is not suitable in this work, since some of the opinions are neither disfavorable nor unfavorable. For example, the odor classification as “undefined” or the color classification as “indifferent” is neither beneficial nor detrimental for the commercialization of the products. Radar-gram graphs are quite often used for sensorial analysis, since they are easy of being observed and interpreted by the readers.
- It would help to explain and offer a hypothesis about the origin of the VOCs of the unloaded emulsions – is there a possible contamination?
Answer: Thanks for this comment. The explanation of some VOC in the unloaded formulation could be explained by the presence of such compounds in the raw materials/excipients used for the formulations preparation. We gave such explanation in the results section.
- Conclusion. Clear and well written, the English is better than at the Intro section.
Reviewer 2 Report
line 61 - better to say "presents a selection of commercial" because it is not possible to collect all the products in the market
line 211 - the word "accopled" ? should be "coupled"
line 247 and others units should be correctly printed "goil" should be "g oil" the same for others units
line 299 and subsequent the reference to the table 4A and 4B is confused it is suggested to split the description and place it under each part (picture and graphs)
line 316 correction "of"
Conclusion - Due to the many variables in cosmetic products and marketing mix it is suggested to evaluate if useful to add some statement regarding this aspect.
Interesting paper with a lot of details and analytical work
Author Response
- line 61 - better to say "presents a selection of commercial" because it is not possible to collect all the products in the market
Answer: We appreciate your comment. This sentence was removed since Table 1 was also removed according to reviewer 1 suggestion.
- line 211 - the word "accopled" ? should be "coupled"
Answer: The correction was made
- line 247 and others units should be correctly printed "goil" should be "g oil" the same for others units
Answer: The corrections were made
- line 299 and subsequent the reference to the table 4A and 4B is confused it is suggested to split the description and place it under each part (picture and graphs)
Answer: Thanks for your observation. Figure 4 was fixed in order to not split the content.
- line 316 correction "of"
Answer: The correction was made
- Conclusion - Due to the many variables in cosmetic products and marketing mix it is suggested to evaluate if useful to add some statement regarding this aspect.
Answer: A statement on this regard was inserted in the conclusion.
Before modifications:
Supercritical CO2 extraction of tomato waste was carried out at 500 bar and 60 °C with an overall extraction yield of ca. 91 % of total oil present in tomato waste.
After modifications:
Cosmetics industry has an increasing interest in the incorporation of natural complex mixtures such as plant extracts, which may impact the color and odor of the final products.
In this work, LEE were extracted from tomato waste and incorporated in preparations for topical application. Supercritical CO2 extraction of tomato waste was carried out at 500 bar and 60 °C with an overall extraction yield of ca. 91 % of total oil present in tomato waste.
- Interesting paper with a lot of details and analytical work
Reviewer 3 Report
The topic and some results of the work are interesting but the paper should be published after significant changes.
The main reasons for recommending a major revision of the paper are methodological limitations for the sensory analysis.
In a acceptance test, a large numbers of subjects (at least 50-100) are required, if valid conclusions are to be drawn. Moreover, more details on the methodology should be provided. Which method was applied? Which scale? Which location of the test? How was the questionnaire built? Why were texture attributes not taken into account?...
In my opinion, it is necessary to specify that these are preliminary sensory results conducted on a small scale (25 subjects). Therefore, significant changes on the paper organization, more focused on physico-chemical characterization (and not on sensory analysis), should be made.
Revisions should include significant changes to the organization and exposition of the paper (e.g. new title and abstract, add chemical data, reorganize materials, summarize results, rewrite conclusions).
Author Response
- The topic and some results of the work are interesting but the paper should be published after significant changes.
- The main reasons for recommending a major revision of the paper are methodological limitations for the sensory analysis.
- In a acceptance test, a large numbers of subjects (at least 50-100) are required, if valid conclusions are to be drawn.
Answer: The reviewer has a point. However, due to COVID-19 pandemic recruiting volunteers was more difficult, as long as volunteers had to smell the tested products. This number of volunteers is not so unusual in biometric assays or for assaying organoleptic characteristics in healthy volunteers (doi:10.3390/cosmetics5020026; doi:10.1016/j.taap.2018.01.018, doi:10.1590/S1516-89132013000200005, DOI:10.19277/BBR.10.1.56).
- Moreover, more details on the methodology should be provided.
Answer: More details were given accordingly.
- Which method was applied?
Answer: Twenty-five healthy untrained testers were assigned to assess different formulations, performing a test of the organoleptic characteristics (odor and color) through a questionnaire. The study protocol was designed by the researchers and the procedures followed were in accordance with the ethical standards of the FFULisboa committee.
- Which scale?
Answer: The scale is described in the methods section “Sensorial analysis of formulations”:
The questions and the scale are as described: the odor intensity (without odor, slightly perceptible, perceptible, very perceptible, intense odor); with exception of volunteers that answered without odor, the volunteers were questioned about odor perception (pleasant and fresh odor, pleasant and hot odor, unpleasant, very unpleasant and undefined); lastly, the color of all formulations were also assessed (unpleasant, slightly pleasant, indifferent, pleasant, very pleasant). The sensory analysis also enabled to study the acceptability of odor and color of each formulation by the volunteers. This parameter was measured by the probability of buying different cosmetic products (face cream, body cream, shampoo and toothpaste) with the odor and color of the respective formulation (would never buy, unlikely, likely, quite likely, purchased the product). At the beginning the volunteers performed the sensorial analysis without knowing any information about the composition of the formulations; after being informed (AI) that LEE-ME and LEE-Macro contained a tomato extract with antioxidant properties, the volunteers answered the same questionnaire about the organoleptic properties and the probability of buying different cosmetic products (acceptability) with the odor and color of the respective formulation (being designated in this work as LEE-ME (AI) and LEE-Macro (AI)). Knowing that information, the sensorial analysis of these two formulations was reassessed.
- Which location of the test?
Answer: All subjects remained in a room free from strong odors, maintained at 25 ± 1° C and 40–60% relative humidity, under natural day light, for 5 min prior to evaluation. The samples in test (at room temperature) were placed in front of each volunteer aligned with the sequential questions of the questionnaire.
- How was the questionnaire built?
Answer: The questionnaire was built electronically as a Google form questionnaire.
- Why were texture attributes not taken into account?...
Answer: Our first intention was to test how LEE odor affects formulation attributes, such as skin feel, tackiness and spreadability. However, due to COVID-19 pandemic recruiting volunteers was more difficult, and most of them refuse to test formulation texture attributes.
- In my opinion, it is necessary to specify that these are preliminary sensory results conducted on a small scale (25 subjects). Therefore, significant changes on the paper organization, more focused on physico-chemical characterization (and not on sensory analysis), should be made.
Answer: Based on reviewer suggestion we made changes on the manuscript focusing the findings more on the physico-chemical characterization and not on sensory analysis.
- Revisions should include significant changes to the organization and exposition of the paper (e.g. new title and abstract, add chemical data, reorganize materials, summarize results, rewrite conclusions).
Answer: Revisions on the title, abstract, results and conclusions were made.
Round 2
Reviewer 3 Report
Authors partially improved the quality of the work. However, some conceptual errors are still present and, in my opinion, not admissible for a publication in this journal.